# Immunogenicity and Safety of an Intradermal BNT162b2 mRNA Vaccine Booster after Two Doses of Inactivated SARS-CoV-2 Vaccine in Healthy Population

**DOI:** 10.3390/vaccines9121375

**Published:** 2021-11-23

**Authors:** Porntip Intapiboon, Purilap Seepathomnarong, Jomkwan Ongarj, Smonrapat Surasombatpattana, Supattra Uppanisakorn, Surakameth Mahasirimongkol, Waritta Sawaengdee, Supaporn Phumiamorn, Sompong Sapsutthipas, Pasuree Sangsupawanich, Sarunyou Chusri, Nawamin Pinpathomrat

**Affiliations:** 1Department of Internal Medicine, Faculty of Medicine, Prince of Songkla University, Songkhla 90110, Thailand; iporntip@medicine.psu.ac.th (P.I.); sarunyouchusri@hotmail.com (S.C.); 2Department of Biomedical Sciences and Biomedical Engineering, Faculty of Medicine, Prince of Songkla University, Songkhla 90110, Thailand; 6310320007@psu.ac.th (P.S.); 5910210489@psu.ac.th (J.O.); 3Department of Pathology, Faculty of Medicine, Prince of Songkla University, Songkhla 90110, Thailand; pornapat.s@psu.ac.th; 4Clinical Research Center, Faculty of Medicine, Prince of Songkla University, Songkhla 90110, Thailand; ssupattr@medicine.psu.ac.th (S.U.); pasuree.s@psu.ac.th (P.S.); 5Department of Medical Science, Ministry of Public Health, Nonthaburi 11000, Thailand; surakameth.m@dmsc.mail.go.th (S.M.); waritta.s@dmsc.mail.go.th (W.S.); 6Institute of Biological Products, Department of Medical Sciences, Ministry of Public Health, Nonthaburi 11000, Thailand; supaporn.p@dmsc.mail.go.th (S.P.); sompong.s@dmsc.mail.go.th (S.S.)

**Keywords:** intradermal, mRNA vaccine, inactivated SARS-CoV-2, COVID-19, immunogenicity

## Abstract

Effective vaccine coverage is urgently needed to tackle the COVID-19 pandemic. Inactivated vaccines have been introduced in many countries for emergency usage, but have only provided limited protection. Heterologous vaccination is a promising strategy to maximise vaccine immunogenicity. Here, we conducted a phase I, randomised control trial to observe the safety and immunogenicity after an intradermal boost, using a fractional dosage (1:5) of BNT162b2 mRNA vaccine in healthy participants in Songkhla, Thailand. In total, 91 volunteers who had been administered with two doses of inactivated SARS-CoV-2 (CoronaVac) were recruited into the study, and then randomised (1:1:1) to received different regimens of the third dose. An intramuscular booster with a full dose of BNT162b2 was included as a conventional control, and a half dose group was included as reciprocal comparator. Both, immediate and delayed adverse events following immunisation (AEFI) were monitored. Humoral and cellular immune responses were examined to observe the booster effects. The intradermal booster provided significantly fewer systemic side effects, from 70% down to 19.4% (*p* < 0.001); however, they were comparable to local reactions with the conventional intramuscular booster. In the intradermal group after receiving only one fifth of the conventional dosage, serum Anti-RBD IgG was halved compared to the full dose of an intramuscular injection. However, the neutralising function against the Delta strain remained intact. T cell responses were also less effective in the intradermal group compared to the intramuscular booster. Together, the intradermal booster, using a fractional dose of BNT162b2, can reduce systemic reactions and provides a good level and function of antibody responses compared to the conventional booster. This favourable intradermal boosting strategy provides a suitable alternative for vaccines and effective vaccine management to increase the coverage during the vaccine shortage.

## 1. Introduction

Severe acute respiratory syndrome coronavirus 2 (SARS-CoV-2) results in COVID-19. Since the end of October 2021, this serious illness has infected more than 242 million patients and has been the cause of over 5 million deaths worldwide. Additionally, this pandemic has consumed vast medical resources and has greatly affected the global economy. Adequate vaccine coverage seems to be the only solution in stopping this pandemic [1]. However, true, mass immunisation has proven to be a real challenge for vaccine developers, policy makers/regulators and principle investigators [2]. Emergency use of various vaccines has been granted on the balance of safety and efficacy of vaccines and vaccine regimens [3].

Recent viral mutations have caused many variants of concern (VOCs) and this has become problematic; for example, the new emerging B.1.617.2 (Delta) strain is now one of the main variants found globally, including in Thailand [4]. A significant mutation at the receptor-binding protein (RBD) increases the viral capacity for replication and transmission [5]. Additionally, the reports of COVID-19 infections among vaccinated individuals, including health care workers, are on the rise [6]. Host immunological responses and the vaccines effectiveness against this variant are limited, which makes it difficult to achieve satisfactory vaccine management during this breakthrough.

Recent studies have shown the efficacy of an mRNA-based vaccine (BNT162b2), and a replication-deficient simian adenovirus vector (ChAdOx1 nCoV-19) against the Delta variant of SARS-CoV-2. The effectiveness of completed ChAdOx1 nCoV-19 vaccination was 74.5%, while the BNT162b2 vaccination was 93.7% [7]. Interestingly, the predicted neutralisation of BNT162b2 against the Delta variant was reduced by 5.8 times compared to the original strain [8]. On the other hand, the neutralising activity in vaccinated individuals with two dosages of CoronaVac (Sinovac) was lower compared to convalescent people [9]. However, the efficacy data of an inactivated SARS-CoV-2 vaccine (CoronaVac) against the Delta strain is still insufficient.

Several reports have shown that the longevity of vaccine inducted immune responses has continued to decline [10,11,12]. Therefore, the existing immunity provided by previous, conventional vaccinations was not as effective against potentially emerging variants of concern [11,13,14]. Recently, heterologous vaccination strategies have been introduced to improve the vaccine immunogenicity and efficacy during a shortage of supply [15]. Vaccine prioritisation and mass administration have been proven to be unsuccessful in the countries with limited vaccine supplies, and have constricted vaccine platforms [15]. Therefore, this current study focused on boosting immunity in addition to the existing inactivated SARS-CoV-2 (CoronaVac) vaccine regimen.

Based on insufficient data on a boosting dosage, the lower volume of vaccine dosages was preferred, so as to minimise dose-dependent adverse reactions. Intradermal administration was then suggested as an appropriate route for introducing reciprocal doses. The immunogenicity and efficacy of fractional intradermal vaccination in comparison with full dose immunisation has been examined in many pathogens, such as influenza virus, rabies virus, poliovirus (PV), hepatitis B virus (HBV) and hepatitis A virus (HAV) [16,17]. In TB vaccines, a prime with an attenuated pathogen, followed by an intradermal boost using a viral vector vaccine, provided superior protection and inducted strong cellular and humoral immune responses [18,19]. Intradermal vaccination of Chimpanzee adenoviral (ChAd) vectored vaccines has also been performed. Intradermal vaccination with a viral vector Malarial vaccine, AdCh63 ME-TRAP, induced comparable immune responses with intramuscular full dose vaccination [20]. Recent studies have suggested that intradermal vaccination with reciprocal doses (1:5 and 1:10) of mRNA-1273 (Moderna) provided similar antibody responses compared to a full dose of conventional intramuscular injections [21]. Therefore, an mRNA-based vaccine, BNT162b2, a lipid nanoparticle–formulated, membrane-anchored SARS-CoV-2 full-length spike protein, was selected to be examined as a booster dose in our study.

Herein, we conducted the safety and immunogenicity study in healthy adults who had received a conventional two dosage of inactivated SARS-CoV-2 vaccine (CoronaVac) with an intradermal booster, using a reciprocal dosage of BNT162b2.

## 2. Results

### 2.1. Study Participants

This study was conducted at the Clinical Research Center, Faculty of Medicine, Prince of Songkla University, Songkhla, Thailand. Demographics of the study participants are shown in Table 1.

Healthy adults, aged 18–60 years (*n* = 91), having received a two dosage-regimen of inactivated SARS-CoV-2 vaccine of more than 8 weeks were recruited into this study. The median of participant age was 39.9 years old, with no differences between the treated groups. The interval between first and second dose of the inactivated vaccine was 21 days, and median time to the booster dose (third dose) was 73 days. The CONSORT diagram is shown in Figure 1. In total, 91 participants were randomised to receive either 0.3 mL of BNT162b2 intramuscularly (group 1, *n* = 30) or a half dosage (0.15 mL) of the mRNA vaccine intramuscularly (group 2, *n* = 30). For the intradermal group, 31 participants received one fifth of the mRNA vaccine intradermally (group 3, *n* = 31). Blood samples were collected on the day of vaccination, and then 14 and 28 days after vaccination for immunological analysis.

### 2.2. Immediate and Delay Adverse Events

The adverse event analysis is illustrated in Table 2. There was only one patient in the intradermal group that developed an immediate adverse event, defined as a local reaction. Regarding delayed, local reactions, 83.5% had at least one reaction, which was non-significant between groups (Figure 2A). The reported pain was common in the intramuscular groups, while the prevalence of swelling, erythema and palpable nodules were significantly increased in the intradermal group (51.6%, 87.1%, and 51.6%%, respectively) (Figure 2B). Regarding delayed systemic reactions, nearly half of the patients had at least one systemic reaction. Interestingly, we observed a significantly lower rate of systemic reactions in the intradermal group (19.4%) compared with 40–70% in the intramuscular groups (*p* < 0.001) (Figure 3A). Fever, headache and myalgia were significantly common in the intramuscular groups (Figure 3B). The treatment data showed most patients in the intradermal group were in grade 1, while the patients in the intramuscular groups needed supportive treatment (Figure 3B). However, the medication needed was not significant different between groups. At 4 weeks post booster, no serious adverse effects were observed.

### 2.3. SARS-CoV-2 Anti-RBD Antibody Responses Induced by Intradermal and Intramuscular BNT162b2 Booster

Anti-RBD IgG levels have been purposed as an immune correlate of protection against SARS-CoV-2 infection [22]. For antibody analysis, blood was drawn and processed. Serum samples were analysed to examine the antibody responses before and after the booster. The level of anti-RBD IgG was measured before boosting in participants who had completed two doses of the inactivated vaccine. After 14 days of receiving the intramuscular booster, the antibody responses were significantly increased in sera of volunteers boosted with a full dose of the mRNA vaccines compared to the non-boosted group (median = 3884 and 52 BAU/mL, respectively, *p* < 0.0001) (Figure 4A). The reciprocal dose of the intramuscular mRNA booster slightly enhanced fewer antibody responses (median = 2837 BAU/mL) compared to the conventional booster (*p* = 0.0505) (Figure 4A). The intradermal booster using one fifth of the standard dosage enhanced half of the IgG responses (median = 1962 BAU/mL) obtained from conventional boosting. After 28 days of receiving the booster, the antigen-specific IgG remained significant in all the boosted groups and was higher compared to the baseline responses (*p* < 0.0001) (Figure 4A). The levels of the antibody responses 4 weeks after boosting were slightly decreased compared to the earlier timepoint, but the differences were not statistically significant (Figure 4A). The antibody responses on day 28 followed the same pattern with day 14. The standard dose boosted group remained superior in enhancing IgG responses compared to the half dose group (median = 2622 and 1952 BAU/mL, respectively), but the difference was approaching significance (*p* = 0.0616) (Figure 4A). Again, the fractional intradermal booster enhanced significantly fewer antibody responses (median = 1205 BAU/mL) compared to the conventional intramuscular boosted group (*p* < 0.0001) (Figure 4A).

### 2.4. Neutralising Antibody against the Delta Variant after Being Boosted with BNT162b2; Intradermally and Intramuscularly

Neutralising antibodies have been showed to correlate with the protectivity of vaccines against the SARS-CoV-2 variants of concern [8,9]. The serum samples were serially diluted before being tested with the live virus. The ability to reduce the infectivity of the virus by 50 percent was presented in this study. More than 8 weeks after two doses of the inactivated vaccines, the neutralising of antibodies against the Delta strain was observed (Figure 4B). Reciprocal antibody titres of the SV SV baseline were less than the titre obtained from convalescent sera from previously infected individuals with Delta strains and other 2020 strains (Figure 4B). After 14 days of receiving the booster with the mRNA vaccine, the neutralising antibody activity was significantly improved in all the boosted groups compared to the baseline (*p* < 0.0001) (Figure 4B). Conventional intramuscular boosters provided significantly higher neutralisation against the Delta variant compared to the fractional intramuscular booster (*p* = 0.0058) (Figure 4B). The neutralising titres in the intradermal group were not significantly different compared to the intramuscular controls. However, there was a trend for lower neutralising activity when boosted intradermally with a one fifth dose of the mRNA vaccine compared to the full dose boosted intramuscularly (*p* = 0.1272) (Figure 4B).

### 2.5. T Cell Responses Induced by Intradermal and Intramuscular BNT162b2 Boosters

While antibody responses provide neutralising functions against the virus entering the body and host cells, cellular responses, especially T cells, play a crucial role in eliminating the virally infected cells [23]. Herein, we analysed PBMCs collected before and after the mRNA vaccine booster. The cells were stimulated ex vivo with S1 peptide pools. The IFN-γ producing cells were then stained and counted to be measured. T cell responses were observed in volunteers previously immunised with inactivated SARS-CoV-2 vaccines (Figure 5). After 14 days of boosting, the IFN-γ secreting T cells were significantly increased after intramuscular boost with both conventional and fractional doses of the mRNA vaccine compared to the baseline (*p* = 0.0139 and 0.0282, respectively) (Figure 5). Reducing the dose by half did not affect T cell responses when boosted intramuscularly (Figure 5). The fractional intradermal boost failed to enhance the T cell responses, showing no difference between pre and post boosting. Significant differences were observed between the booster routes. The intramuscular boost with a full dose or half dose of the mRNA vaccine provided superior levels of IFN-γ SFCs compared to the intradermal route (*p* = 0.0305 and 0.0589) (Figure 5).

## 3. Methods

### 3.1. Study Procedures

This study was registered in the Thai Clinical Trials Registry (TCTR20211004001). Before enrolment, all participants provided written informed consent. The approval was obtained from the Human Research Ethics Committee (REC. 64–368–4–1). The trial was conducted according to the principles of Good Clinical Practice.

### 3.2. Immediate and Delayed Adverse Events

Immediate local and systemic adverse events were monitored for 30 min after injection. Local reactions were measured as millimeter of wheal and flare, and systemic reactions were observed by vital signs recorded after finishing the 30-minute observation. The delayed adverse events were monitored at seven days and 4-weeks after the booster. The participants were retrieved from telephone-based interviews, by experienced research nurses at seven days, and they completed a questionnaire regarding adverse events at 4 weeks after the booster. Delayed adverse events were categorised into either: local and systemic reactions. The severity of the reaction was graded into 3 levels. No medication required was grade 1, medication required was grade 2 and a doctors’ attention required was grade 3. The rates of each adverse reaction are reported in this study.

### 3.3. Sample Processing

Blood samples were collected on the day of vaccination and then 14 and 28 days after the 3rd dose. Blood samples were obtained and divided between a clotted blood tube and a heparinised tube. Samples were processed within 4–6 h of the blood draw. The clotted blood samples were processed for serum collection. The tubes were centrifuged at 1800 r.p.m. for 10 min, and the serum was harvested for storage at −80 °C until required. The heparinised blood tubes were processed for the collection of PBMCs and plasma, by density gradient centrifugation. Blood from the same participant was pooled into a 50 mL conical centrifuge tube and spun to separate blood plasma. The plasma was then collected and stored at −80 °C. The remaining blood samples were diluted with RPMI (Gibco) and laid into SepMATE tube containing Lymphoprep (STEMCELL Technologies). The samples were then centrifuged at 1200× *g* for 10 min with the brake on. The top layer was poured into a fresh 50 mL tube and topped up with RPMI, then spun 300 g for 8 min. The cell pellet was washed again with RPMI. After the last wash, the cell pellet was resuspended in 3 mL of R10 media (RPMI-1640 containing 1% penicillin–streptomycin, 2 mM L-glutamine and 10% fetal calf serum (FCS, Labtech) for counting. Cells were diluted in Trypan blue, and counted using a counting chamber for use in fresh assays or for cryopreservation. All remaining cells were centrifuged (300× *g* for 8 min) and adjusted into a concentration of 3 × 10^6^ PBMCs per mL in freezing media (FCS containing 10% DMSO). The cell suspensions were aliquoted and transferred to CoolCells (Corning) for freezing at −80 °C overnight. Tubes were then transferred into liquid nitrogen storage until required.

### 3.4. Quantification of SARS-CoV-2 Anti-S RBD Antibodies

The level of immunoglobulin G (IgG) to the receptor binding domain (RBD) of S1 subunit spike protein of SARS-CoV-2 were measured and quantified in serum samples by using ARCHITECT i System (Abbott, Abbott Park, IL, USA) chemiluminescent microparticle immunoassay (CMIA) (SARS-CoV-2 IgG II Quant, Abbott Ireland, Sligo, Ireland), with a measuring reportable range from 6.8 Abbott Arbitrary Unit (AU/mL) to 80,000.0 AU/mL. Values higher than 50 AU/mL were considered positive. Regarding the unit, WHO suggests the WHO binding antibody unit (WHO BAU/mL) to be used with the SARS-CoV-2 IgG II Quant assay. The correlation between relationships of the AU/mL unit to the WHO BAU/mL unit is at 0.142 × AU/mL, with a 0.999 correlation coefficient.

### 3.5. Plaque Reduction Neutralisation Test (PRNT)

In this study, PRNT was performed by the Institute of Biological Products’; a WHO- contracted laboratory at the Department of Medical Sciences. Vero cells were seeded at 2 × 10^5^ cells/well/3 mL and placed in a 37 °C, 5% CO_2_ incubator for 1 day. Test sera were initially diluted at 1:10, 1:40, 1:160 and 1:640, respectively. SARS-CoV-2 virus was diluted in culture medium to yield 40–120 plaques/well in the virus control wells. Cell control wells, convalescent patient serum samples were also included as assay controls. The neutralisation was performed by mixing the equal volume of diluted serum and the optimal plaque numbers of SARS CoV-2 virus at 37 °C in water bath for 1 h. After removing the culture medium from Vero cell culture plates, 200 ul of the virus-serum antibody mixture were inoculated into monolayer cells and then rocked with the culture plates every 15 min for 1 h. Three mL of overlay semisolid medium (containing 1% of carboxymethylcellulose, Sigma Aldrich, St. Louis, MO, USA, with 1% of 10,000 units/mL Penicilin-10,000 ug/mL Streptomycin (Sigma, Tucson, AZ, USA) and 10% FBS) were replaced after removing excessive viruses. All plates were incubated at 37 °C, 5% CO_2_ for 7 days. Cells were fixed with 10% (*v*/*v*) formaldehyde, then stained with 0.5% crystal violet in PBS. The number of plaques formed was counted in triplicate wells, and the percentage of plaque reduction at 50% (PRNT50) was calculated. The PRNT50 titre of test samples is defined as the reciprocal of the highest test serum dilution for which the virus infectivity is reduced by 50% when compared with the average plaque counts of the virus control. This was calculated by using a four-point linear regression method.

### 3.6. Ex vivo IFN-γ ELISpot Assays

ELISpot assays were performed on freshly isolated PBMCs before and after the booster dose. The MultiScreen-IP Filter plates (Millipore, Burlington, MA, USA) were coated with 10 μg/mL of human anti-IFN-γ coating antibodies (clone 1-D1K, Mabtech, Sweden) in a carbonate buffer (Sigma-Aldrich, Burlington, MA, USA), and stored at 4 °C overnight. The coated plates were washed three times with PBS and blocked with R10 media for at least 1 h at 37 °C. After the blocking, 2.5 × 10^5^ PBMCs were added into assigned wells along with the S1 SARS-CoV-2 peptide pool. A total of 134 peptides were made, as 15-mers overlapping by 10 amino acids (ProImmune, Oxford, UK) and used at a final concentration of 2 µg/mL. Each assay was performed in duplicate and incubated for 16–18 h at 37 °C with 5% CO_2_. Plates were developed by washing six times with PBS/T, followed by the addition of 1 μg/mL of anti-IFN-γ detector antibody (7-B6-1-Biotin, Mabtech) to each well. After a 2-h incubation, plates were washed again, and 1:1000 SA-ALP was added for 1 h at RT. After a final wash step, plates were developed using BCIP NBT-plus chromogenic substrate (Mabtech). ELISpot plates were counted using an Immunospot Microanalyzer

(Cellular Technology Limited). Responses were averaged across duplicate wells, and the mean response of the unstimulated (negative control) wells were subtracted. Results are shown as SFCs/10^6^PBMCs.

### 3.7. Statistical Analysis

Statistical analyses were completed using GraphPad Prism 9 software (GraphPad Software Inc., San Diego, CA, USA). To define the statistical significance, the Mann–Whitney test was used to compare two groups, while the Kruskal–Wallis followed by Dunn’s multiple comparisons test, were tested when analysing multiple groups. Values of a *p* ≤ 0.05 were considered as statistically significant. * *p* ≤ 0.05, ** *p* ≤ 0.01, *** *p* ≤ 0.001, **** *p* ≤ 0.0001, ns = non significance.

## 4. Discussion

This study highlights the differences in humoral and cellular immune responses between heterologous vaccination with a conventional intramuscular booster and fractional intradermal booster of the mRNA vaccine (BNT162b2) from healthy volunteers having been administered with two doses of the inactivated SARS-CoV-2 vaccine. Additionally, systemic adverse reactions among those receiving an intradermal fractional dose of BNT162b2 were significantly decreased compared to those receiving an intramuscular conventional dose of BNT162b2. Unfavorable local reactions were not commonly observed among those receiving both routes of the BNT162b2 booster.

Intradermal injection impressively minimized the systemic adverse reactions, which have been indicated in previous studies, including ours [20,21]. The self-limited local reactions have been observed in the vaccination route, and shown to be dose dependent. In this study, even though the fractional dose using one fifth of BNT162b2 mRNA vaccine provided fewer antibody responses compared to the conventional dose and route, minimal favourable local and systemic side effects were seen. This suggests that there is a possibility for dose escalation, titrating the dosage to reach expected immunity levels within acceptable adverse reactions. Moreover, the results of this study may provide an alternative vaccine administration route in people who develop systemic reactions from prior injections, leading to the situation of vaccine reluctance.

This study showed the decrease of antibody responses after 8 weeks post vaccination among people who had received two doses of an inactivated SARS-CoV-2 vaccine compared to previous studies [24,25]. The immunological responses of the participants were markedly low in both antibody and T cell responses. Without a booster dose, these participants were at risk for SARS-CoV-2 infection, and severe clinical outcomes [26]. The decline of vaccine efficacy to prevent infection along with severity indicated an urgent need of the heterologous boosting concepts, following the conventional two doses of the inactivated SARS-CoV-2 vaccine. In many countries, including Thailand, most people were vaccinated with two doses of an inactivated SARS-CoV-2 vaccine (CoronaVac). The recent outbreak could be explained by the waning immunity after an inadequate vaccination program [27]. Our findings provide alternative vaccine management to increase vaccine coverage and boosted host immunity against new, emerging viral strains.

The shortage of COVID-19 vaccine has been reported in several countries [28]. To increase vaccine distribution and to achieve heard immunity, most of the vaccines have been prioritised to unvaccinated people, which makes it difficult to obtain vaccines for boosting [29]. Our findings purposed a reasonable solution by using less vaccine volume, combined with an intradermal injection. The immunogenicity in this study was consistent with previous reports on intradermal coronavirus vaccines in various platforms. Interestingly, a fractional dose of mRNA-1273 (Moderna) provided comparable antibody responses with that of a conventional dose and route of vaccination [21,30,31]. However, our results suggest that antibody levels obtained from the BNT162b2 booster appeared to be dose dependent. This could be explained by the actual dose contained in each vaccine type. The mRNA-1273 vaccine contain higher doses (100 μg) compared to the BNT162b2 mRNA vaccine (30 μg), or the mRNA-1273, which is just simply more immunogenic compared to another mRNA vaccine [32,33].

Humoral and cellular responses after receiving a fractional dose of the intradermal boost were inconsistent with several vaccine studies on other viral infections, using different vaccine platforms [34,35]. Intradermal vaccination provided comparable responses compared to intramuscular injection in adeno viral vector vaccines against COVID-19 and malaria (Pinpathomrat et al. unpublished data) [20]. Interestingly, we found that BNT162b2 enhanced strong immune responses when it was delivered intramuscularly but the responses were halved when injected intradermally. A possible explanation is the abundance of dendritic cells that could be infected by viral vectors, and work as an antigen presenting cell at the lymph nodes, so as to augment T helper 2 axis, which activate B cells to produce antigen-specific antibodies [36]. However, when the mRNA vaccine was injected intradermally, dendritic cells could not effectively uptake the vaccine materials, resulting in less immune reactions/responses when compared to intramuscular injection, which enhanced T helper 1 responses. However, local dendritic cells at the injection site can cause a local reaction among those receiving a fractional intradermal vaccine [20,21]. The explanation for the less systemic reaction among those receiving an intradermal vaccine is unclear, while the proposed explanation is for dose-dependent associated systemic reactions [20,21]. However, no study has evaluated the immunogenicity and reactogenicity of intradermal BNT162b2 (Pfizer) as a first dose.

Antigen-specific antibodies and neutralising antibody levels have been purposed as immune correlations of protection against SARS-CoV-2. Higher levels of antibodies have been observed in highly protective vaccines, such as mRNA and viral vector vaccines [22,26]. Therefore, anti-RBD-IgG were used for measuring the primary outcome of this study, which has shown no differences between the conventional intramuscular booster and afractional intradermal booster. The neutralising assays performed in pre-clinical and phase I studies of the current vaccines were tested against the wild-type strains [37,38,39,40]. However, the vaccine efficacy has been reported to be decreased during the breakthrough of the recently mutated virus [7]. Neutralising activity against new variants of live viruses is the closest method to predict a vaccines performance [9,26]. In the sera of vaccinated participants, the antibody neutralising function against the Delta strain was significantly improved after both intradermal and intramuscular boosting. Without boosting, the neutralising function was poor, which is consistent with previous reports showing low neutralising activity after completion of two doses of the inactivated vaccine [9]. The protective efficacy of the intradermal booster is still being evaluated in a larger population.

T cell responses are crucial to evaluate immunogenicity of the vaccines, especially in pre-clinical studies and phase I trials [37,38,39,40,41,42]. The T cell analyses are usually different in each study, but IFN-γ producing T cells were favourable to observe for cellular responses in vaccine trails [37,38,43]. This study observed a higher response of IFN-γ secreted T cells after boosting with the intramuscular mRNA vaccine booster compared to intradermal injection. This is consistent with previous studies on T cell responses after being vaccinated with two doses of intramuscular BNT162b2 [44]; however, no study has shown T cell responses after intradermal vaccination with this mRNA vaccine. Comparable T cell responses between intradermal and intramuscular injection were observed after immunisation with ChAd63 viral vector vaccine expressing malarial antigens as well as in our unpublished data on intradermal ChAdOx1 nCoV-19 [20]. Comprehensive T cell studies are still needed concerning immune correlates of protection, and to understand the difference between intramuscular and intradermal vaccination.

Several limitations should be acknowledged. First, our study provides reassurance of BNT162b2 (Pfizer) vaccine tolerability and safety, wherein most reactions were mild and transient. However, the specific, serious adverse events, such as myocarditis or intradermal adverse reactions, as in skin necrosis, were not reported due to the small number of participants.

Due to the relatively short duration of the study, protectivity against the infection and disease severity was not possible to access. The study focused on a previously vaccinated population, with inactivated SARS-CoV-2 (Sinovac); therefore, it is limited in applying the findings to other vaccine platforms. BNT162b2 (Pfizer) was the only booster vaccine examined in this study. Hence, applications of other mRNA vaccines or other vaccine strategies remain unclear. The interval of more than 8 weeks between the completed vaccination and the booster was enrolled in this study, which means its application is limited for the shorter intervals.

Further data supporting the ongoing evaluation of the intradermal booster of the BNT162b2 vaccine, in a larger population, so as to observe rare reactogenicity and evaluate the booster efficacy is required.

## 5. Conclusions

A phase I clinical trial of an intradermal BNT162b2 mRNA booster, in healthy volunteers who completed two dosages of an inactivated SARS-CoV-2 vaccine was conducted in Songklanagarind hospital, Thailand. Local and systemic reactions as well as antibody and T cell responses of this heterogenous vaccination were evaluated. It was found that a fractional dose (one fifth) of the BNT162b2 mRNA vaccine, administered intradermally, can greatly reduce systemic reactions compared to a full dose of an intramuscular booster. Interestingly, immune responses obtained from the reciprocal boosting were high, but less than the conventional booster with a full dose vaccine. Even though the antibody levels after intramuscular injection of the mRNA vaccine were superior compared to the fractional intradermal boosting, the neutralising function against the Delta variant of SARS-CoV-2 seemed to be comparable between these two routes of vaccination. T cell responses were also observed following the same trend with the serology data.

Our results have potential for significant impact during the breakthrough of the Delta strain in Thailand, and in many other countries. The inactivated SARS-CoV-2 vaccines were initially introduced, which now require a booster. While the vaccine coverage is still not effective enough to generate herd immunity; reciprocal boosting using the intradermal route is a crucial tool to enhance immunity with fewer vaccine amounts, and more importantly, fewer side effects.

## Figures and Tables

**Figure 1 vaccines-09-01375-f001:**
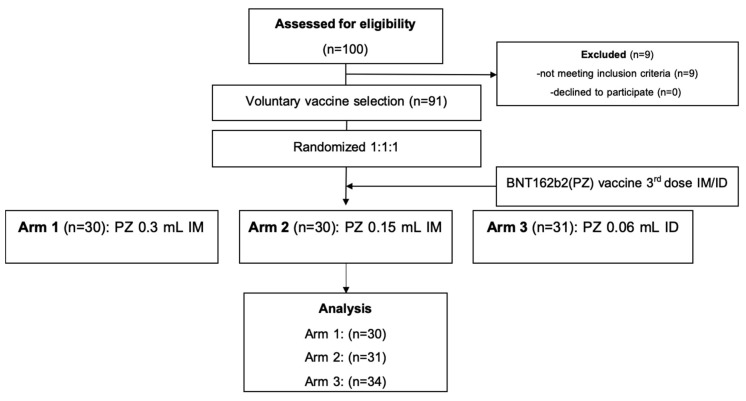
CONSORT chart. Abbreviations: intradermal (ID); intramuscular (IM).

**Figure 2 vaccines-09-01375-f002:**
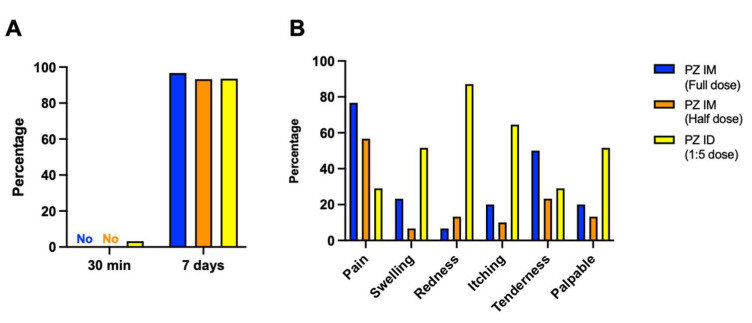
Solicited local adverse reactions at 30 min and 7 days after boosting. Full dose (blue) and half dose (orange) of intramuscular mRNA booster, after completed vaccination with two doses of inactivated SARS-CoV-2. Some of the previously vaccinated individuals were boosted with a fractional dose of intradermal mRNA vaccine (yellow). (**A**) The immediate and delayed local reactions were observed after injection. (**B**) Seven days after boosting, local adverse events were recorded for comparing between booster groups.

**Figure 3 vaccines-09-01375-f003:**
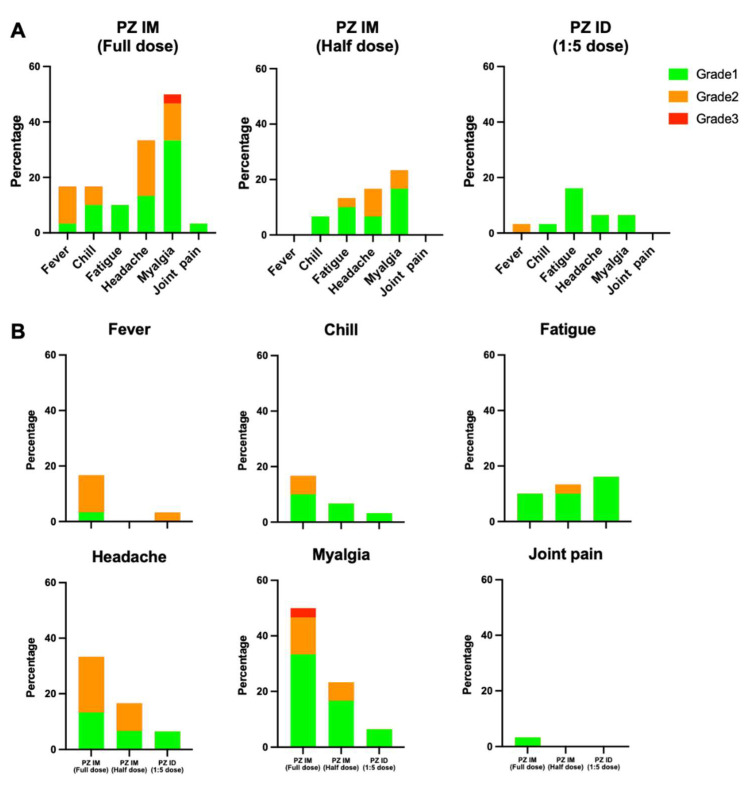
Solicited systemic adverse reactions at 7 days after boosting. Full dose and half dose of the mRNA vaccine were given intramuscularly after two doses of inactivated SARS-CoV-2 vaccines (PZ IM, full dose and half dose). One-fifth of the standard mRNA vaccine dose was delivered intradermally (PZ ID, 1:5 dose). (**A**) The systemic adverse events were graded, as per medical needs, and presented in percentage. The self-limited systemic reactions were grade 1 (green). The reactions requiring medications were grade 2 (orange), in need of medical attention were grade 3 (red). (**B**) Systemic reactions were presented separately to compare between the three vaccinated groups.

**Figure 4 vaccines-09-01375-f004:**
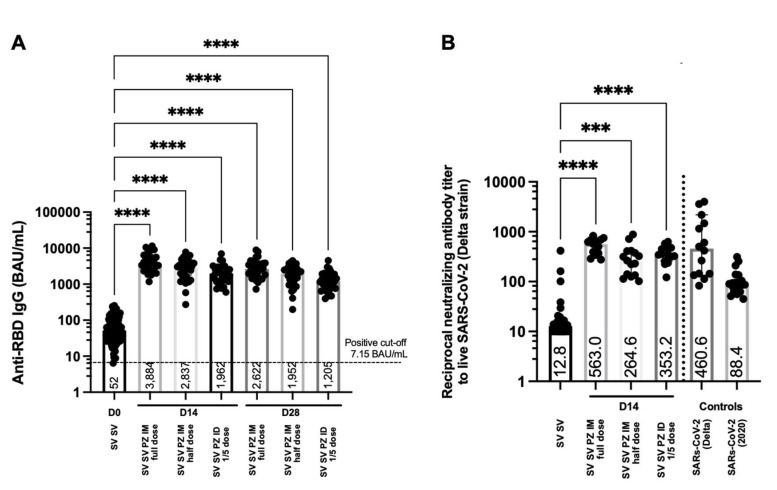
Antibody responses and neutralising function after boosting. Participants who had received two doses of inactivated SARS-CoV-2 vaccines were recruited in this study (SV SV). Conventional (full dose) and fractional dose (half dose) of the mRNA vaccine were administered intramuscularly (SV SV PZ IM). The fractional dose (1:5 dose) of the mRNA boost was delivered intradermally (SV SV PZ ID). Blood was collected pre (D0) and post boosting 14, 28 days (D14, D28). (**A**) Serum samples were analysed using CMIA to measure anti-RBD IgG. (**B**) Neutralising antibodies against the Delta variant was tested using PRNT. Convalescent sera were included as controls. Each symbol represents one participant, and the number is the median of each group (*n* = 30–31 volunteers). Statistical significance was determined using the Kruskal–Wallis test, with Dunn’s multiple comparisons test. *** *p* ≤ 0.001; **** *p* ≤ 0.0001.

**Figure 5 vaccines-09-01375-f005:**
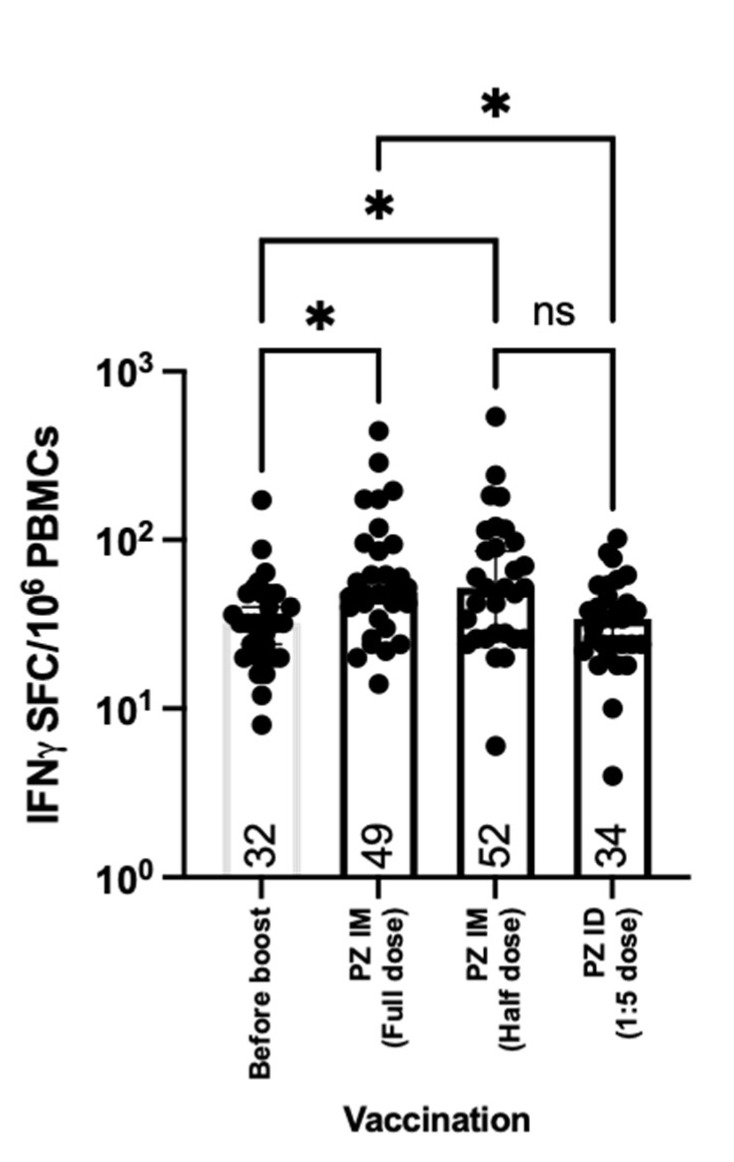
T cell responses after boosting. All volunteers were previously vaccinated with two doses of inactivated SARS-CoV-2 vaccines (SV SV). The fractional dose (1/5 dose) of the mRNA vaccine was injected intradermally, as a booster dose (SV SV PZ ID). Full doses and half doses of the mRNA vaccine were immunised intramuscularly as a standard route of injection (SV SV PZ IM). Blood was drawn before (D0) and after the booster dose for 14 days (D14). The blood samples were processed to obtain PBMCs. The fresh PBMCs were stimulated with S1 peptide pools, before measuring IFN-γ secreted cells, using ELISpot. Each symbol represents one participant, and the number is the median of each group (*n* = 30–31 volunteers). Statistical significance was determined using the Kruskal–Wallis test, with Dunn’s multiple comparisons test. *, *p* ≤ 0.05; ns = non significance.

**Table 1 vaccines-09-01375-t001:** Baseline characteristic of the intramuscular (IM) and intradermal (ID) BNT162b2 (PZ) mRNA vaccine booster in general population.

Baseline Characteristics	Total	PZ IM Full	PZ IM Half	PZ ID	*p* Value
	*n* = 91 (%)	*n* = 30 (%)	*n* = 30 (%)	*n* = 31 (%)	
Gender					
Female	51 (56.0)	22 (73.3)	13 (43.3)	16 (51.6)	0.054
Male	40 (44.0)	8 (26.7)	17 (56.7)	15 (48.4)	
Mean age, y (SD)	39.9	40.8 (9.2)	40.6 (8.1)	38.4 (9)	0.7
Vaccine duration, days (IQR)	21	21 (21,21.8)	21 (21,21.8)	21 (21,22.5)	0.751
Time to booster, days (IQR)	73	73 (69.8,76)	73 (68,74)	73 (69,74)	0.888

**Table 2 vaccines-09-01375-t002:** Adverse events of the intramuscular (IM) and intradermal (ID) boosting, with BNT162b2 (PZ) in the healthy, general population.

	Total	PZ IM Full	PZ IM Half	PZ ID	*p* Value
Characters	*n* = 91 (%)	*n* = 30 (%)	*n* = 30 (%)	*n* = 31 (%)	
Immediate reaction	1 (1.1)	0 (0)	0 (0)	1 (3.2)	1
Delay reaction					
Local reactions	76 (83.5)	27 (90)	21 (70)	28 (90.3)	0.078
Pain	49 (53.8)	23 (76.7)	17 (56.7)	9 (29)	<0.001
Swelling	25 (27.5)	7 (23.3)	2 (6.7)	16 (51.6)	<0.001
Erythema	33 (36.3)	2 (6.7)	4 (13.3)	27 (87.1)	<0.001
Nodule	26 (28.6)	6 (20)	4 (13.3)	16 (51.6)	0.002
Systemic reactions	39 (42.9)	21 (70)	12 (40)	6 (19.4)	<0.001
Fever	6 (6.6)	5 (16.7)	0 (0)	1 (3.2)	0.029
Chill	8 (8.8)	5 (16.7)	2 (6.7)	1 (3.2)	0.165
Fatigue	12 (13.2)	3 (10)	4 (13.3)	5 (16.1)	0.925
Headache	17 (18.7)	10 (33.3)	5 (16.7)	2 (6.5)	0.025
Myalgia	24 (26.4)	15 (50)	7 (23.3)	2 (6.5)	<0.001
Treatment (*n* = 39)					0.253
Grade 1	21(53.8)	9 (42.9)	7 (58.3)	5 (83.3)	
Grade 2	18 (46.2)	12 (57.1)	5 (41.7)	1 (16.7)	

## Data Availability

The data that supports the findings of this study are available from the corresponding author upon reasonable request.

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
