# Peer review of "Immunogenicity and Safety of an Intradermal BNT162b2 mRNA Vaccine Booster after Two Doses of Inactivated SARS-CoV-2 Vaccine in Healthy Population"

_vaccines, 2021, doi:10.3390/vaccines9121375_

Round 1

Reviewer 1 Report

In this manuscript, Porntip et al evaluated the immunogenicity and safety of an intradermal BNT162b2 mRNA vaccine boost after two doses of inactivated SARS-CoV-2 vaccine in healthy population. The authors performed a phase I randomized control trial to observe safety and immunogenicity after an intradermal boost using 1/5 dosage of BNT16b2 mRNA vaccine in healthy participants. 91 volunteers who had completed 2 doses of inactivated SARS-CoV-2 (CoronaVac) were recruited in the studies and randomized (1:1:1) to received different regimen of the third dose. Intramuscular boost with full dose of BNT162b2 was included as controls and half dose group was completed as reciprocal comparators. The immediate and delay side effects and humoral and cellular immune responses were examined. The results showed that intradermal boost group display significantly less systemic side effects and comparable immune response. Taken together, the ID boost using 1/5 dosage mRNA vaccine can reduces systemic reactions and provides good antibody responses compared with conventional boost. This paper has great guidance effort for providing a alternative for vaccinees and effective vaccine management to increase the coverage during the vaccine shortage, especially PZ mRNA vaccine.

Minor comments:

1) the tile change to “Immunogenicity and safety of an intradermal BNT162b2 mRNA vaccine booster after two doses of inactivated SARS-CoV-2 vaccine in healthy population

2) table 1 should also include male participants data.

Author Response

Point 1: the tile change to “Immunogenicity and safety of an intradermal BNT162b2 mRNA vaccine booster after two doses of inactivated SARS-CoV-2 vaccine in healthy population

Response 1: We thank the reviewer for helpful suggestions. We have changed the title as suggested.

Point 2: table 1 should also include male participants data.

Response 2: We agreed with the reviewer. We have included male participant data in the table 1 as suggested.

Reviewer 2 Report

In this study the authors compare different doses and routes of administration of a BNT162b2 mRNA booster shot in a phase I clinical trial to observe safety and immunogenicity. The cohort consists of healthy patients who have previously received two doses of the CoronaVac vaccine. The data demonstrates that a 1/5 booster dose could induce both a humoral and cellular immune response when given intramuscularly, but only induced humoral responses when given intradermally.  However, neither response was as high as the regular intramuscular dose. While side effects with the intramuscular dose were significant (but tolerable), the intradermal dose had reduced systemic side effects.

Overall, a really interesting, timely and useful study with a good rationale, and a good discussion and comparison to other studies. I just have a couple of comments below.

Line 59 - I would specify which route you would recommend, i.e. intradermal or intramuscular.

Line 140 - Please include the demographic and site of your phase I trial here and in your abstract - i.e. city and country.

Author Response

Point 1: Line 59 - I would specify which route you would recommend, i.e. intradermal or intramuscular.

Response 1: We thank the reviewer for helpful suggestion. As suggested, We have specified that the Intradermal route is more favourable providing a good alternative for vaccinees and effective vaccine management.

Point 2: Line 140 - Please include the demographic and site of your phase I trial here and in your abstract - i.e. city and country.

Response 2: We agreed with the reviewer. The demographic and the study site were added in the result section as well as in the abstract.